

# Do subterranean mammals use the Earth's magnetic field as a heading indicator to dig straight tunnels?

Sandra Malewski[1], Sabine Begall[1,2], Cristian E. Schleich[3],
C. Daniel Antenucci[3] and Hynek Burda[1,2]

[1] Department of General Zoology, University of Duisburg-Essen, Essen, Germany
[2] Department of Game Management and Wildlife Biology, Czech University of Agriculture, Prague, Czech Republic
[3] Laboratorio de Ecología Fisiológica y del Comportamiento, Instituto de Investigaciones Marinas y Costeras (IIMyC), Consejo Nacional de Investigaciones Científicas y Técnicas (CONICET), Universidad Nacional de Mar del Plata, Mar del Plata, Buenos Aires, Argentina

## ABSTRACT

Subterranean rodents are able to dig long straight tunnels. Keeping the course of such "runways" is important in the context of optimal foraging strategies and natal or mating dispersal. These tunnels are built in the course of a long time, and in social species, by several animals. Although the ability to keep the course of digging has already been described in the 1950s, its proximate mechanism could still not be satisfactorily explained. Here, we analyzed the directional orientation of 68 burrow systems in five subterranean rodent species (*Fukomys anselli*, *F. mechowii*, *Heliophobius argenteocinereus*, *Spalax galili*, and *Ctenomys talarum*) on the base of detailed maps of burrow systems charted within the framework of other studies and provided to us. The directional orientation of the vast majority of all evaluated burrow systems on the individual level (94%) showed a significant deviation from a random distribution. The second order statistics (averaging mean vectors of all the studied burrow systems of a respective species) revealed significant deviations from random distribution with a prevalence of north–south (*H. argenteocinereus*), NNW–SSE (*C. talarum*), and NE–SW (*Fukomys* mole-rats) oriented tunnels. Burrow systems of *S. galili* were randomly oriented. We suggest that the Earth's magnetic field acts as a common heading indicator, facilitating to keep the course of digging. This study provides a field test and further evidence for magnetoreception and its biological meaning in subterranean mammals. Furthermore, it lays the foundation for future field experiments.

Corresponding author
Hynek Burda,
hynek.burda@uni-due.de

## INTRODUCTION

Across the globe, about 250 rodent species, belonging to different unrelated taxa (six families, 38 genera), have convergently adapted to permanent life in self-excavated extensive underground burrow systems (for review see *Begall, Burda & Schleich, 2007*; *Lacey, 2000*; *Nevo, 1999*). Burrow systems inhabited by single individuals (solitary species) or families (social species) can reach lengths of several hundred meters or, in social

species, even kilometers (*Brett, 1991*; *Šklíba et al., 2012*; *Šumbera et al., 2003*, *2008*, *2012*). Here, the animals live, communicate, orientate in space and time in a dark, monotonous sensory environment free of most orientation cues which are available aboveground. Especially light propagation is highly limited in subterranean burrow systems, so that vision is of little use for subterranean rodents there (*Kott et al., 2014*). Since environmental cues are mostly lacking in the subterranean ecotope, rodents are forced to rely on idiothetic cues to orientate which are, however, prone to accumulations of errors. Consequently, an external reference that could be used for orientation and navigation would be of high value (*Moritz et al., 2007*). While the question, how these animals can orientate in their complex underground maze, has been repeatedly addressed (*Burda, 1987*; *Eloff, 1951*; *Kimchi, Etienne & Terkel, 2004*; *Kimchi & Terkel, 2001*), one interesting sensory ecological aspect remained understudied. How do subterranean mammals manage to keep the course of digging? This question was first raised approximately 70 years ago in 1951 when Eloff mentioned the "remarkable ability (of the South-African *Cryptomys* mole-rat) to follow a direction or find a spot where it previously bored a tunnel" (*Eloff, 1951*, p. 145). The African mole-rat's ability to maintain its course while digging long, straight tunnels (*Eloff, 1951*) gave rise to speculations on possible orientation cues: air currents (*De Graaff, 1972*; *Eloff, 1958*; *Poduschka, 1978*) and acoustic cues (*Eloff, 1951*; *Müller & Burda, 1989*; *Rosevear, 1969*) have been discussed, as well as internal mechanisms based on kinesthetic and/or vestibular cues (*Teroni et al., 1988*). However, none of these mechanisms provide a satisfactory explanation for the highly efficient directional orientation observed in *Cryptomys hottentotus* (*Eloff, 1951*).

Inspired by the mentioned enigmatic ability, about 40 years after Eloff, *Burda (1987)* postulated that the mole-rats might use magnetic cues to orientate and navigate, an ability which was then explicitly proven in the laboratory repeatedly, specifically for the Zambian Ansell's mole-rat (*Fukomys anselli*, formerly assumed to represent the same species as the South African *C. hottentotus* and in papers published before 2006 named so) (*Burda, 1987*; *Burda et al., 1990*; *Malewski et al., 2018*; *Marhold, 1997*; *Thalau et al., 2006*; *Wegner, Begall & Burda, 2006*). The phenomenon of magnetoreception, meaning an animal's sensory ability to extract information from the Earth's magnetic field, has been investigated since the 1960s (for review see *Begall, Burda & Malkemper, 2014*; *Wiltschko & Wiltschko, 1995*). However, studies with the purpose to unravel the function of the mammalian magnetic sense are still rare (*Holland et al., 2006*; *Kimchi, Etienne & Terkel, 2004*). Following the finding that African mole-rats are magnetosensitive (*Burda et al., 1990*), *Lovegrove, Körtner & Körtner (1992)* investigated directional orientation of burrow systems of Damaraland mole-rats (*F. damarensis*) in the field, inferring that the "orientation of the burrow system with respect to a specific compass orientation may represent an intrinsic requirement of successful geomagnetic orientation and direction finding" (*Lovegrove, Körtner & Körtner, 1992*, p. 631). However, no correlation between the Earth's magnetic field and the burrow system's orientation was found. Years later, *Schleich & Antinuchi (2004)* performed a comparable study on burrow systems of Talas tuco-tucos (*Ctenomys talarum*), and achieved also negative results. However, in both

studies, the data were analyzed by means of a Chi$^2$-test, which does not take the specific characteristics of circular data into account (e.g., $10°$ is close to $350°$), and which is therefore less powerful than circular statistics. Furthermore, the previously used "polygon method" expressed directional alignment of the burrow system through a line connecting the most distant points of a polygon surrounding the burrow system. Since this method ignores all the shorter side branches of a burrow system within the circumferential polygon, a new, more detailed and accurate method would be of avail.

In our study, we revive the question raised by *Eloff (1951)*, and test if subterranean rodents use the Earth's magnetic field as a heading indicator to keep a straight course of digging. Keeping a digging direction, for example, during foraging (*Burda, 1987*) or during natal or mating dispersal (*Hickman, 1990*; *Nevo, 1999*; *Rado, Wollberg & Terkel, 1992*) would be advantageous, as digging curvy tunnels would imply increased or even devastating energetic costs; note that costs for digging are between 360 and up to 3,400 times higher than if the animal moves the same distance above ground (*Vleck, 1979*). We may assume that straight tunnels longer than, for example, one m are not the product of a single digging bout, and in the case of social species, not even of a single individual (*Jarvis et al., 1994*). This fact points out the necessity of a heading indicator to keep the course of digging. While diverse heading indicators (visual or olfactory landmarks, sun position, wind direction) can be used to keep the course of locomotion aboveground, availability of such cues is restricted underground.

To test the hypothesis of a magnetic heading indicator, we analyzed the directional orientation of complete burrow systems excavated in the field of five subterranean rodent species, three of which are already known to be magnetosensitive (*Oliveriusová et al., 2012*): two *Fukomys* mole-rat species (*F. anselli* and *F. mechowii*), the silvery mole-rat (*Heliophobius argenteocinereus*), the blind mole rat (*Spalax galili*), and the Talas tuco-tuco (*C. talarum*) by means of two different methods.

## MATERIAL AND METHODS

Our dataset consisted of 68 burrow systems of five subterranean rodent species (*F. anselli*, *F. mechowii*, *H. argenteocinereus*, *C. talarum*, *S. galili*), belonging to three nonrelated families with different geographic distribution, and representing both solitary or social lifestyles (Table 1). All burrow systems were mapped within the framework of previous field ecological studies (*Lövy et al., 2015*; *Schleich & Antinuchi, 2004*; *Šklíba, Šumbera & Chitaukali, 2010*; *Šklíba et al., 2009*, *2012*; *Šumbera et al., 2003*, *2008*, *2012*), and the maps were provided to us for this study. They were digitized true to scale by means of Quantum GIS (QGIS Geographic Information System, www.qgis.osgeo.org) for burrow systems of silvery mole-rats, *Fukomys* mole-rats, and *Spalax*, and by means of ImageJ (*Schneider, Rasband & Eliceiri, 2012*) for burrow systems of Talas tuco-tucos. We pooled the data of both species of *Fukomys* mole-rats in order to increase the sample size. It should be noted that both species are phylogenetically related, live in the same habitat in the same geographic region and have the same lifestyle. The direction of magnetic North was noted as accompanying information as usual in geographical mapping. The data has been collected blindly with respect to the analysis of compass

**Table 1 Tested rodent species (sorted alphabetically by their common names).**

| Species | Family | Social/ solitary | Number of burrow systems | Locality | Year of excavation | Reference, burrow systems were originally used for |
|---------|--------|------------------|--------------------------|----------|--------------------|-----------------------------------------------------|
| Ansell's mole-rat (*Fukomys anselli*) | Bathyergidae | Social | 7 | Lusaka East Forest Reserve (Zambia) | 2010 | *Šklíba et al. (2012)* |
| Giant mole-rat (*Fukomys mechowii*) | Bathyergidae | Social | 2 | Ndola Hill Forest Reserve (Zambia) | 2009 | *Šumbera et al. (2012)* |
| Silvery mole-rat (*Heliophobius argenteocinereus*) | Bathyergidae | Solitary | 31 | Blantyre, Mulanje, Mpalaganga (Malawi) | 2000, 2005 | *Šklíba et al. (2009)*, *Šklíba, Šumbera & Chitaukali (2010)*, and *Šumbera et al. (2003, 2008)*, and additional previously unpublished burrow systems (cf. raw data) |
| Talas tuco-tuco (*Ctenomys talarum*) | Ctenomyidae | Solitary | 19 | Mar de Cobo, Buenos Aires (Argentina) | 1988, 1989 | *Schleich & Antinuchi (2004)* |
| Upper Galilee Mountains blind mole rat (*Spalax galili*) | Spalacidae | Solitary | 9 | Upper Galilee Mountains (Israel) | 2012 | *Lövy et al. (2015)* |

**Note:**
Number of mapped burrow systems, locality and year of excavation, as well as references to the studies describing burrow architecture of the respective species are listed.

directionality, since the original purpose of the excavation was not to study burrow orientation, but to obtain data about other aspects of subterranean life (e.g., burrow architecture). In our study, the systems were digitally analyzed twice, by applying two different methods: the "polygon method," which was applied already in previous studies (*Lovegrove, Körtner & Körtner, 1992*; *Schleich & Antinuchi, 2004*), and the "long tunnel method," introduced by us here (Fig. S1). Since the animals' original digging direction (unidirectional) cannot be deduced from the excavations or drawings, bidirectional analysis was the method of choice, that is, data are doubled (modulo 360) before being analyzed, and the resulting mean vector is then back-converted, thus ranging in the interval (0°; 180°). All axial values are reported as XX°/XX° (N/S).

The "polygon method" (Figs. S1A and S1C) is based on drawing a convex polygon connecting the outermost points of a burrow system. A diagonal, marking the longest distance within the polygon, was drawn, representing the main axis of the burrow system. The direction of the longest distance within the polygon was measured blindly (i.e., without knowing the direction of magnetic North) by means of a digital compass rose. Subsequently, all measured values were unblinded by normalizing the data relative to the direction of true magnetic North.

The "long tunnel method" (Figs. S1B and S1E) weights the magnetic axial direction of the branches within a burrow system depending on their length. The underlying hypothesis concerning this approach was the following: The longer the distance, which was dug in a specific magnetic direction, the stronger is the animal's preference for that particular direction. To apply this method, first, we determined the directions of all straight tunnels of a burrow system blindly (i.e., the evaluator did not know the direction of magnetic North). We defined a "straight tunnel" as straight segment including small turns/curves which ended at a clear change in direction of more than 30° (see Fig. S1E).

Only straight tunnels that were at least as long as 5% of the polygon's circumference were included in our data set. The tunnels' axial directions were measured by means of a digital compass rose. After unblinding the data, vector analysis of each straight tunnel direction weighted by the tunnels' lengths was performed to calculate the mean direction for each burrow system separately. Therefore, the vector's direction was treated as value and its length as frequency (see frequency editor of Oriana 4.0, Kovach computing systems).

All drawings of polygons and diagonals as well as their evaluations were performed with Fiji (*Schindelin et al., 2012*). Circular statistics (*Batschelet, 1981*) and vector analyses were calculated by using Oriana 4.0 (Kovach computing systems). Axial mean values were calculated for each species and each method (for the "long tunnel method," the mean vector (grand mean) was calculated over the mean axial directions of the respective tunnels). The Rayleigh-test of uniformity was used to test for significant deviations from a random distribution. Watson–Williams *F*-test (pairwise comparison) was applied to compare the burrow systems' directions achieved by both methods for each species, respectively.

## RESULTS

By applying the "long tunnel method," the axial orientation of the vast majority of all evaluated burrow systems (94%) showed a significant deviation from a random distribution, that is, in 64 out of the 68 analyzed tunnel systems the straight tunnel segments were oriented significantly in a certain axial compass direction (*Fukomys*: 100%; *H. argenteocinereus*: 90%; *C. talarum*: 95%; *S. galili*: 100%). Considering the second order statistics (averaging the mean vectors of all analyzed burrow systems of a respective genus or species), a significant deviation from random orientation was found for the *Fukomys* mole-rats, the silvery mole-rat, and the Talas tuco-tuco (Fig. 1). An axial directional preference for the North–South axis was observed for the silvery mole-rat (Fig. 1A), while it was slightly shifted to NNW/SSE in the Talas tuco-tuco (Fig. 1B). The *Fukomys* mole-rats showed a preference for the NE/SW (Fig. 1C). No prevailing magnetic direction of the studied burrows was found in the blind mole rat (*S. galili*) (Fig. 1D).

Since most burrow systems ($N = 31$) were available for the silvery mole-rats, we analyzed whether the tunnel systems' lengths and locality of excavation (Blantyre, Mpalaganga, Mulanje; Table 1) had an effect on the orientation of the mean axial direction. A significant deviation from random orientation was found for systems which were shorter than 100 m ($N = 15$, $\mu = 1/181°$, SD = 31°, $P = 0.007$; mean length = 57 m, SD = 26 m), in contrast to systems which were longer than 100 m ($N = 16$, $\mu = 15/195°$, SD = 48°, $P = 0.406$; mean length = 192 m, SD = 68 m). With respect to the locality of excavation, a significant deviation from random orientation was only observable in one out of three localities (Blantyre: $N = 17$, $\mu = 5/185°$, SD = 34°, $P = 0.015$; Mpalaganga: $N = 10$, $\mu = 169/349°$, SD = 52°, $P = 0.702$; Mulanje: $N = 4$, $\mu = 20/200°$, SD = 30°, $P = 0.272$). Furthermore, we did not find statistical sex-dependent differences in mean orientation for any of the solitary species (statistics not shown). The "polygon method" revealed that the alignment of the burrow systems of the Talas tuco-tuco deviated significantly from random orientation (Fig. 1F). A similar strong tendency was

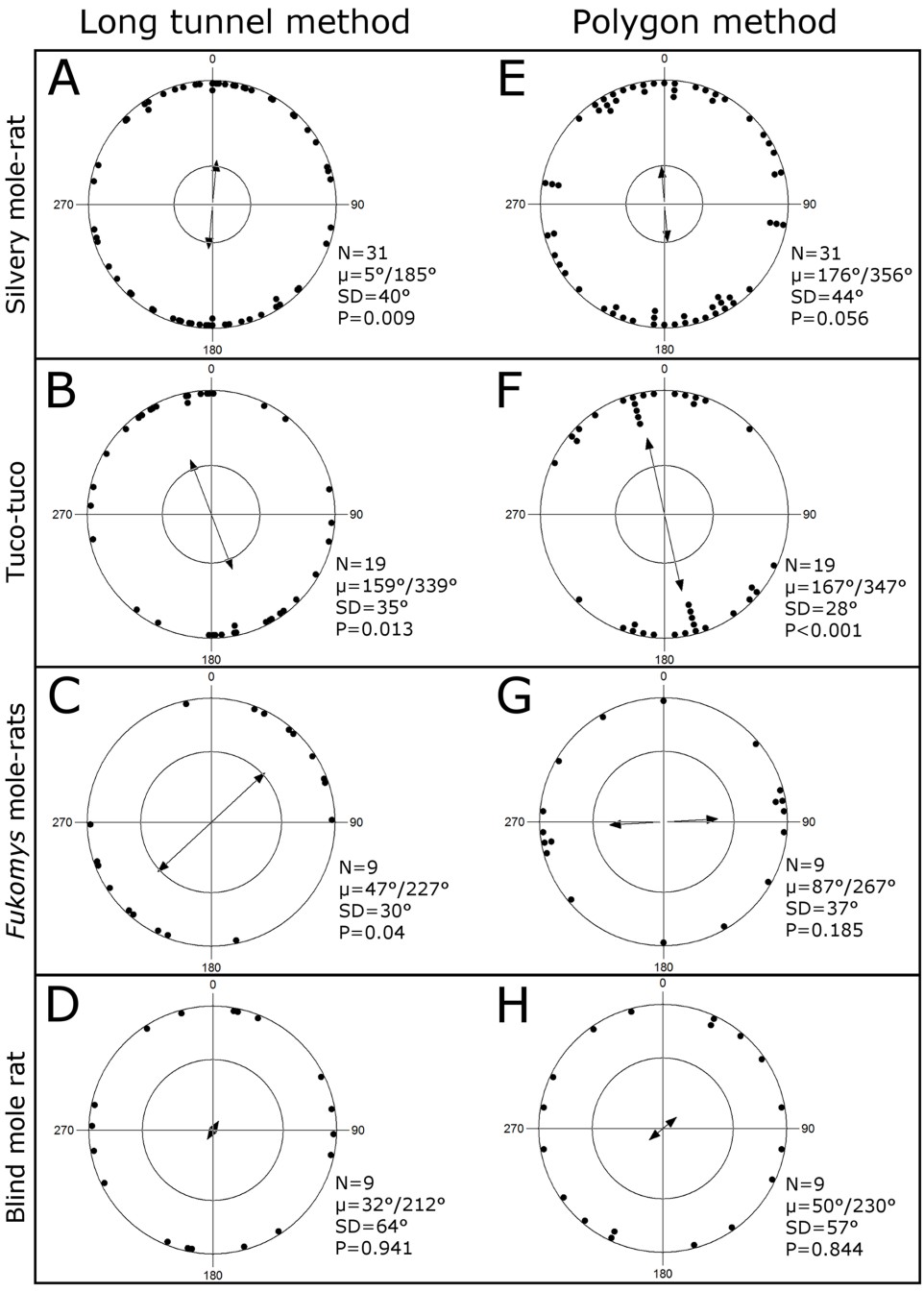

**Figure 1 Burrow system's orientation.** Directional orientation of the burrow systems (estimated by the "long tunnel method," A–D) and the prevailing direction of burrows (estimated by the "polygon method," E–H) of the tested rodent species—the silvery mole-rat *H. argenteocinereus* (A, E), Talas tuco-tuco *C. talarum* (B, F), *Fukomys* mole-rats Ansell's mole-rat *F. anselli* and the giant mole-rat *F. mechowii* (C, G), and the Upper Galilee Mountains blind mole rat *S. galili* (D, H) (sorted by sample size)—relative to magnetic North (0°). Two mirrored dots represent the axial direction of one burrow system, which is in case of the long tunnel method an axial mean vector calculated over all directions of straight tunnel segments weight by the segments' lengths. The double-headed arrow indicates the (grand) mean vector, and the inner circle marks the significance level of 0.05 (Rayleigh test). Sample size (N), mean axial direction (μ), circular standard deviation (SD), and *P*-value are given for each tested species.

observable for the silvery mole-rat's systems (Fig. 1E). The burrow systems of the *Fukomys* mole-rats (Fig. 1G) and the blind mole rat (Fig. 1H) showed random directional orientation using the "polygon method".

The axial directional orientation of the burrow systems as assessed by the "polygon method" (Figs. 1A–1D) and the prevailing direction of the burrows estimated by the "long tunnel method" (Figs. 1E–1H) did not differ significantly except for the *Fukomys* mole-rats (Watson–Williams *F*-test (pairwise comparisons for each species): *Fukomys* mole-rats: $P = 0.042$, silvery mole-rat: $P = 0.423$, tuco-tuco: $P = 0.455$, blind mole rat: $P = 0.636$), however, a greater scatter was observable by applying the "polygon method."

## DISCUSSION

Figure 1 illustrates that the "polygon method" provides less consistent results than the "long tunnel method." Since the "polygon method" has been employed in previous studies (*Lovegrove, Körtner & Körtner, 1992*; *Schleich & Antinuchi, 2004*), we decided to use this method and present its results for the sake of comparability, too. However, the "polygon method" is generally suitable only for rather linear instead of circular burrow systems, where the long diagonal within a polygon represents the directional alignment of the whole burrow system. In a more circular system, several diagonals of similar lengths can be drawn (Fig. S1D). Various ecological factors influence the geometry of a burrow system. Radiality (high branching) of a burrow system is expected to increase with uniformity and predictability of food resources, age of the burrow system, and with the number of inhabitants (*Romañach et al., 2005*; *Sichilima et al., 2008*; *Thomas, Swanepoel & Bennett, 2016*). Moreover, the architecture of burrow systems can vary seasonally and in dependence on the inhabitant's sex (e.g., dispersing and mate-seeking individuals might build more linear tunnels) (*Antinuchi & Busch, 1992*; *Nevo, 1999*; *Šklíba et al., 2012*; *Šumbera et al., 2003, 2008, 2012*). Indeed, the index of circularity (formula: $4\pi$ (area)/perimeter², 0 = linear, 1 = circular; *Romañach et al., 2005*) identifies 60% of the studied burrow systems of the solitary silvery mole-rats (Fig. 2A) and 70% of the systems of the solitary tuco-tucos as being rather linear (with an index <0.7 as exemplarily defined threshold separating the upper third as being circular), while all systems of the social *Fukomys* mole-rats exhibited values >0.7 (Fig. 2B).

In contrast to the "polygon method," the "long tunnel method" revealed constant results for different species independent from the respective indices of circularity. Having applied this method, we found a significant deviation from random axial orientation for burrowing longer straight tunnels along a certain axis, which was specific for each burrow system. Moreover, the second order statistics (averaging mean vectors of all burrow systems of a respective genus or species) revealed significant deviation from random orientation in all the tested species but *S. galili* (Fig. 1), including a prevalence of NNW/SSE in case of Talas tuco-tucos and NE–SW in case of *Fukomys* mole-rats. The fact that the studied burrow systems have been uncovered in different regions (even continents) and areas (e.g., grassland, woodland, mountainous areas), and thus accompanying different habitats, as well as in different years (1988–2012; Table 1), support the assumption that the Earth's magnetic field seems to be a globally common and

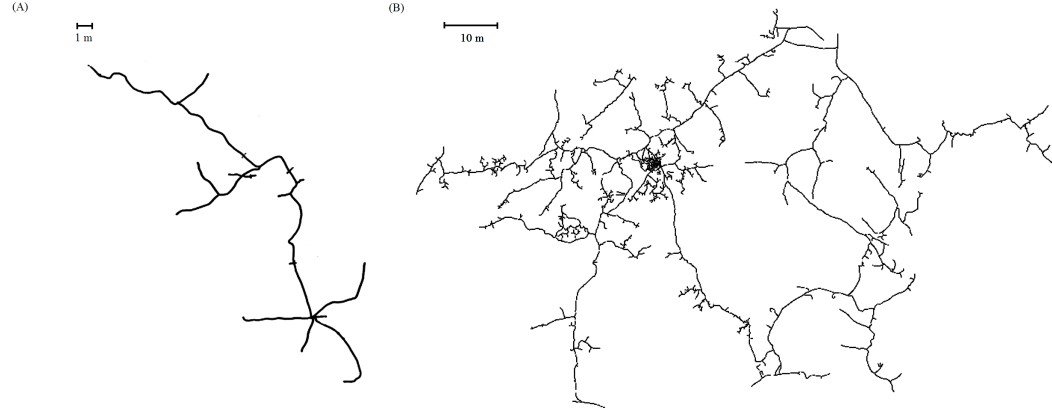

**Figure 2 Examples of burrow systems of a (A) solitary (here *Heliophobius argenteocinereus*) and (B) social species (here *Fukomys anselli*).** Index of circularity (0 = linear, 1 = circular; *Romañach et al., 2005*) for (A) = 0.42, and (B) = 0.83.

stable potential heading indicator. Accordingly, we suggest that the studied species use magnetic cues as heading indicator to keep the course of digging. This hypothesis does not exclude that the animals use also further, locally available heading indicators, for this purpose. Indeed, magnetoreception has been proved in the laboratory for *F. anselli* (*Burda et al., 1990*; *Marhold, Wiltschko & Burda, 1997*), *F. mechowii* (*Oliveriusová et al., 2012*), and *H. argenteocinereus* (*Oliveriusová et al., 2012*). Interestingly, magnetoreception was not proved in laboratory maze experiments in *C. talarum* (*Schleich & Antinuchi, 2004*), which, however, could have been due to an inappropriate experimental design as discussed by the authors themselves (*Schleich & Antinuchi, 2004*). On the other hand, magnetoreception has been proven in *Spalax judaei* (*Kimchi & Terkel, 2001*; *Marhold et al., 2000*), a close relative of *S. galili* for which no directional preference for digging was found here. The reasons, why no directional digging preference was found in *S. galili*, can be manifold. All analyzed burrow systems originate from a single small area in the Upper Galilee Mountains, and six of nine examined burrow systems were located in basaltic soils. Basaltic soils are generally characterized by rather strong magnetic signatures and the studied region is known for magnetic anomalies (*Eppelbaum, Ben-Avraham & Katz, 2004*) that might compromise magnetic orientation. Furthermore, the population density in basaltic soil is five times higher than in rendzina soil (*Lövy et al., 2015*). The avoidance of crossing a neighboring burrow system might influence the burrow systems orientation and thus possibly override orientation via magnetic cues. Therefore, it is of great interest to analyze comparative data on directionality of burrow systems of *Spalax* from other geographic regions and other habitats (cf., *Nevo et al., 1995*; *Reyes, Nevo & Saccone, 2003*; *Savić, 1973*; *Yağci, Coşkun & Aşan, 2010*).

The burrow systems of *Heliophobius* analyzed here (*N* = 31) were excavated at three different localities, characterized by different landscapes (Blantyre, *N* = 17: grassland; Mpalaganga, *N* = 10: woodland; Mulanje, *N* = 4: mountainous) and thus habitats. A significant deviation from random axial orientation was only observable for the systems from Blantyre. The larger scatter of mean vectors at Mulanje might be partially explained

by the lower sample size. Secondly, at Mpalaganga and Mulanje, several systems were excavated in cultivated areas, where higher density and uniformity of food resources might have had influenced foraging strategies and thus also the burrow architecture. Another factor, which seems to affect the burrow system's orientation of silvery mole-rats is the system's length—while shorter systems (<100 m) were oriented significantly in North–South direction (every other direction would have been equally plausible), longer systems (>100 m) were not. This is, however, not surprising, considering that the larger a system becomes, the stronger an original alignment might be overridden or masked. Trying to explain these findings, we have to consider that a system's size presumably grows with its age. Consequently, a system's axial orientation may rather be observable when it is shorter (*in statu nascendi*) compared to when it is longer and older.

While using a magnetic compass as a heading indicator, in general, the animals could follow every direction, and indeed, species-specific differences in chosen nest-building directions were already reported repeatedly (for review see *Oliveriusová et al., 2012*). However, the North–South axis might be the direction of choice. A preference for this axis, as observed in our study for silvery mole-rats and Talas tuco-tucos, is common in studies on magnetic orientation (*Malkemper et al., 2015*; *Oliveriusová et al., 2014*; *Phillips, 1986*) and magnetic alignment (for review see *Begall et al., 2013*; *Malkemper, Painter & Landler, 2016*), and is observable in diverse contexts: landing in waterfowl (*Hart et al., 2013*), escape in roe deer (*Obleser et al., 2016*), activity in cattle and deer (*Begall et al., 2008*), flamingos (*Nováková et al., 2017*), corvids (*Pleskač et al., 2017*), wild boars and warthogs (*Červený et al., 2017*), carps (*Hart et al., 2012*). Considering that species-specific burrowing behavior in North American *Peromyscus* rodents was reported to have a genetic basis (*Dawson, Lake & Schumpert, 1988*; *Hu & Hoekstra, 2017*; *Metz et al., 2017*; *Weber, Peterson & Hoekstra, 2013*), we would like to point to the possibility that "magnetic digging" might be genetically determined, too, representing a highly interesting research question.

Since magnetic and further orientation cues are not exclusive, apart from the Earth's magnetic field, several other cues might influence the alignment of the burrow systems. The most crucial external cues might be food resources (*Heth et al., 2002*; *Romañach et al., 2005*), soil condition (implying the presence of stones and other obstacles) (*Ebensperger & Bozinovic, 2000*; *Kimchi, Reshef & Terkel, 2005*; *Luna, Antinuchi & Busch, 2002*; *Zuri & Terkel, 1997*), and landscape characteristics (e.g., dunes, vegetation, slopes, water currents), and even subsurface geological fractures might function as orientation cues (*Olson & Pollard, 1989*). Presence of food resources influences goal-directed burrowing at shorter distances (*Eloff, 1951*; *Voigt, 2014*). Since geophytes are usually clumped, and can be located at a limited distance by smelling plant kairomones diffused in the soil (*Heth et al., 2002*; *Lange et al., 2005*), an optimal foraging strategy is expected to involve digging straight scouting tunnels with perpendicular shorter branches in fertile areas (following kairomone cues). Although season is expected to affect the digging activity (*Sichilima et al., 2008*) and general architecture of the burrow system (*Šumbera et al., 2003*), it is highly unlikely that its main orientation, if present, changes seasonally, wherefore we do not assume significant seasonal influences.

## CONCLUSION AND OUTLOOK

To conclude, the *Fukomys* mole-rats, the silvery mole-rat, as well as the Talas tuco-tuco were shown to dig their burrow systems with respect to a certain axis, and are assumed to use the Earth's magnetic field as a possible heading indicator. In future studies, animals might be released at a new site so that the genesis of the new burrow system might be monitored in situ as, for example, done by *Voigt (2014)*. Furthermore, digging experiments should be performed under controlled laboratory conditions to unequivocally experimentally demonstrate that subterranean rodents use magnetic cues to keep a heading direction during digging.

## ACKNOWLEDGEMENTS

We thank R. Šumbera and M. Lövy for providing most of the data and for their helpful comments on the manuscript. Further thanks are due to J. Šklíba for offering digital maps of tunnel systems of the Ansell's mole-rats, W. N. Chitaukali, V. Dvořáková, M. Elichová, E. Hrouzková, H. Konvičková, O. Kott, J. Kubová, V. Mazoch, J. Ritter as well as all local helpers who assisted in the field, and N. Oberste for assistance with the analyses.

### Funding

S. Malewski was funded by a PhD fellowship of the German National Academic Foundation (Studienstiftung des deutschen Volkes). H. Burda and S. Begall received support funded by grant "EVA4.0," No. CZ.02.1.01/0.0/0.0/16_019/0000803 financed by OP RDE and H. Burda the Grant Agency of the Czech Republic (project. nr. 15-21840S). We acknowledge support by the Open Access Publication Fund of the University of Duisburg-Essen. The funders had no role in study design, data collection and analysis, decision to publish, or preparation of the manuscript.

### Grant Disclosures

The following grant information was disclosed by the authors:
PhD fellowship of the German National Academic Foundation (Studienstiftung des deutschen Volkes).
"EVA4.0": No. CZ.02.1.01/0.0/0.0/16_019/0000803.
Grant Agency of the Czech Republic (project. nr. 15-21840S).

### Competing Interests

The authors declare that they have no competing interests.

### Author Contributions

- Sandra Malewski conceived and designed the experiments, performed the experiments, analyzed the data, contributed reagents/materials/analysis tools, prepared figures and/or tables, authored or reviewed drafts of the paper, approved the final draft.

- Sabine Begall conceived and designed the experiments, performed the experiments, analyzed the data, contributed reagents/materials/analysis tools, authored or reviewed drafts of the paper, approved the final draft.
- Cristian E. Schleich performed the experiments, analyzed the data, contributed reagents/materials/analysis tools, approved the final draft.
- Carlos D. Antenucci performed the experiments, analyzed the data, contributed reagents/materials/analysis tools, approved the final draft.
- Hynek Burda conceived and designed the experiments, performed the experiments, analyzed the data, contributed reagents/materials/analysis tools, authored or reviewed drafts of the paper, approved the final draft.

## Data Availability

The raw data are provided at https://www.uni-due.de/imperia/md/images/fb10_bio/allgemeine_zoologie/Malewski_et_al._raw_data.xlsx.

## Supplemental Information

Supplemental information for this article can be found online at http://dx.doi.org/10.7717/peerj.5819#supplemental-information.

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
