# Peer review of "Do subterranean mammals use the Earth’s magnetic field as a heading indicator to dig straight tunnels?"

_PeerJ, doi:10.7717/peerj.5819_

## Round 0.1 · original submission · Major Revisions

As can be seen from the reviews, both reviewers are of the view that the manuscript needs to be substantially revised in order to be ready for publication. Much of the work needed is related to the rationale, justification fro design choices, and the conclusions that can be drawn from the data - both are satisfied that the experiment itself was adequately designed and conducted.

·

Basic reporting

This manuscript contains concise, straightforward results and a new analysis method that is a marked improvement from its predecessor. By reworking how the background and technical challenges are presented, these contents will be much more accessible to a broader scientific audience.

1) The authors do a good job in the introduction arguing for why circular statistics is a better choice than prior statistical approaches. They should also provide rationale for their decision to develop a new method for analysis, since it is a major contribution of this work. The ‘long tunnel method’ has an advantage when analyzing circular burrow systems (and conversely the ‘polygon method’ struggles with this), and I think this should be introduced earlier in the manuscript. This will provide the readers with expectations that are satisfied when interpreting the directional orientation plots.
2) The introduction discusses positive magnetoreception findings from laboratory experiments with species that, at first glance (see point #5), are not sampled in the manuscript. Providing some phylogenetic context either in the text or in a figure would greatly benefit a reader that is not a Rodentia expert in understanding why these referenced findings may be applicable to the sampled species.
3) Both points (1) and (2) can be addressed in figure form by providing a phylogenetic tree annotated with species/genus names average circularity scores and then representative burrow system images from a more linear species and more circular species (like in the current Figure 2).
4) What is the rationale behind which species were chosen for analysis? If it is simply availability of data, that is fine, but then I would like some more context regarding this sampled set. Do they represent unrelated taxa? Are their burrow systems representative of the range of fossorial rodents? (size, circularity, etc.)
5) Please clarify which specie(s) is used in the Eloff and early Burda experiments. In the referenced review Begall et al. 2014, an early Burda paper (Burda 1987) is cited with the following comment: “…Ansell’s mole rats (Fukomys anselli, in earlier papers denoted as Cryptomus hottentotus).” This comment is not repeated in this manuscript, such that the referenced paper by Burda appears to be studying species that is not sampled in the paper. However, this review suggests that there HAS been a history of magnetoreception laboratory experiments with F. anselli, which is sampled in the paper. If so, please explicitly state this, as it would strengthen the rationale for sampling Fukomys immensely. Also, the cited article in this manuscript (Burda 1989) and one cited in the review have the same title and journal name, but the review has the article’s publication year as 1987. If this is indeed the same referenced article, please reconcile.
6) The raw data is supplied and clearly organized. It appears the number of significant figures differs between methods. If this is not due to greater accuracy in one technique, please make this consistent across data.
7) Please provide scale bars for Figure 2. There are some small markings and annotations on the burrows in Figure 2A and Figure S1A-B. These markings should either be addressed in the figure legends or removed.
8) The manuscript is clearly written in professional language. A few phrasing and clarification suggestions are listed below.

27 – Replace “in” with “over.” Please specify “long time.” Hours to weeks? Comma between “and” and “in social species.”
28 – Replace “has already been” with “was.”
59 – Replace “during” with “while.”
63 – Replace “neither” with “none.” Remove “seems to be able.”
68 – Replace “also” with “then.”
70 – Remove “e. g.”
73 – Clarify “biological meaning.” Function? Underlying mechanism?
231 – Since Fukomys were pooled, “genus or species.”
246 – Replace “way” with “hand.”
248 – Remove comma.
252 – Replace “what” with “that”
258 – Remove “e. g.”
265 – Remove “e. g.”
275 – Italicize “in statu nascendi.”
295 – Are parentheses necessary?
345 – Typo.

Experimental design

1) For the ‘long tunnel method’, was there a minimum cutoff for the length of tunnel to be included as a “straight tunnel”? Was this a relative or absolute cutoff and how was it determined? Assuming that only tunnels marked with a red line were included in the analysis, it appears many small tunnels in Figure S1E that meet the direction change criterion were not marked. This is necessary information for replication.
2) There should be some justification for pooling the two Fukomys species because congeners can be very different in their burrowing behavior. Do the F. anselli data alone show non-random orientation?

Validity of the findings

Data sets and analysis appear sound, provided that decisions made in experimental design were well justified. The interpretation of the results is fair and the the conclusion and outlook are well stated.

Reviewer 2 ·

Basic reporting

The manuscript by Sandra Malewski provides interesting and original data that significantly contribute to long-lasting debate as to whether subterranean mammals prefer some direction while digging their tunnel systems and what sensory modality guides this behavior. As far I can judge, the authors have used all available published (and also some unpublished) data describing excavated tunnel systems. Another strength of the study that allows certain degree of generalization is inclusion of five species representing distantly related subterranean rodents as well as solitary and social species.
The paper is generally well written; the introduction provides an adequate context; references are adequate and are properly cited.

One think I miss in the Introduction is information that subterranean rodents likely use landmark independent navigation, path integration in their burrow systems. It is dependent exclusively on self-generated, idiothetic cues it is prone to rapid accumulation of errors. That is why subterranean rodents need external directional reference, i.e., a compass sense (for review see Moritz et al 2007).

While it is generally known within community zoologists studying subterranean mammals, it may be useful for a general reader to mention explicitly that light propagation is very limited in burrows and that sense of vision is of little use in underground ecotope.

Experimental design

Methodology, design of the experiment and statistical analyses are adequate. Utilization of circular statistics and “the long tunnel method” constitute important improvements when compared to Chi2 test and the polygon method has been employed in previous studies.

“The long tunnel method” need to be described in greater detail to be reproducible by another investigator. Specifically, exact information of how vector addition was done is needed. What software was utilized?

Validity of the findings

Results are convincing and substantiate majority of the conclusions the authors have drawn. However, there are few statements in the Discussion that are poorly justified or unclear to me:

1) Lines 219–222: “Although the statistics of the examination by means of the ‘polygon method’ points to a significant deviation from a random distribution for the Talas tuco-tuco’s burrow systems, this method is more prone to ecological effects than the ‘long tunnel method’, and thus less suitable to reveal any directional preference.“

The meaning of this statement is unclear and somewhat illogical. Both the long tunnel method and the polygon method yielded very similar directional preference in case of the Talas tuco-tuco’s. In fact, application of the polygon technique resulted in a smaller scatter of data points. This suggest that the polygon method is well applicable and reliable for analysis of mean direction of simple, linear burrow systems of (at least some) solitary species. Why the authors suggest exact opposite?

2) Lines 234–237: “Since the studied burrow systems have been uncovered in different regions (even continents) and areas (e.g. grassland, woodland, mountainous areas), and thus accompanying different habitats, as well as in different years (1998-2012; Table 1), the Earth's magnetic field seems to be the only globally common and stable potential heading indicator.”

This argument is not tenable. The preferred directions differed between studied species and the silvery mole-rats from only one put of three localities exhibited significant directional preference. Therefore, the authors should refrain from argumentation just mentioned. Nevertheless, I agree that magnetic compass sense most likely serve as external directional reference simply because it has been repeatedly shown that they use magnetic compass under laboratory conditions.

3) Line 232: The directional preference of Fukomys mole-rats (47°/227°) should not be described as roughly north-south as it actually is loser to west-east axis.

4) Lines 287-289: ” A preference for the North-South axis could be interpreted as a response in accordance with a magnetic polarity compass, where northward and southward can be distinguished (Thalau et al., 2006), what might be easier than calculating an angle to the poles.”

I do not understand the logic of this argument. Moreover, it is a pure speculation. I strongly recommend its omission.

5) Lines 292-294: “ …… it is not unlikely that the use of the Earth’s magnetic field by subterranean rodents as common heading indicator during digging straight tunnels is genetically determined, too, possibly explaining similar results in different species found here.”

This argument is not convincing. First, the directional preferences of the studied species are not that much similar. Actually, directional preference of Fukomys mole-rats might be significantly different from that of Tuco-tuco. Second, and more importantly, the preferred directions reported here do differ from those reported in laboratory experiments (the silvery mole-rats: field directional preference -north/south, laboratory directional preference – east; Fukomys anselli: field directional preference -north-east/south-west, laboratory directional preference – south-east), the fact that seems to be inconsistent with hard-wired, genetically determined behavior.

Additional comments

Minor comments:
1) Lines 133-134: “..values were re-blinded…”; Line 143: “ After re-blinding the data”
Are you sure about correctness of these expressions?

2) Lines 177-179: “The ‘polygon method’ revealed that the alignment of the burrow systems of the Talas tuco-tuco deviated significantly from random orientation, while a similar strong tendency was observable for the silvery mole-rat’s systems.”

Consider reformulation of this sentence.

3) Line 258: (e.g.) (Nevo et al., 1995, Reyes et al., 2003, Savić, 1973, Yağci et al., 2010) should read (e.g., Nevo et al., 1995, Reyes et al., 2003, Savić, 1973, Yağci et al., 2010)

4) Lines 309-310:” Furthermore, we did not find statistical sex-dependent differences in mean orientation for any of the solitary species.

This information shall be moved to Results.

---

## Round 0.2 · Minor Revisions

There are a few minor suggestions that need to be addressed.

·

Basic reporting

The authors have sufficiently addressed my earlier questions and concerns.

Two final remarks:

If possible, increase the image quality of Figure 2B. The file version I see is quite pixelated and there are grayed areas. A file quality more similar to the complex burrow in the supplemental file would be preferred.

In the raw data, the C. talarum data has some 'long tunnel method' values that include commas. Please check if these are typos. Also, the 'long tunnel method' for Fukomys has less significant figures than the rest. Could this also be a data entry/Excel issue?

Experimental design

No comment.

Validity of the findings

No comment.

Reviewer 2 ·

Basic reporting

no comment

Experimental design

no comment

Validity of the findings

no comment

Additional comments

The revised version of the manuscript is significantly improved. I have only few comments:
1. Lines 54 – 56: “Especially light propagation is highly limited in subterranean burrow systems, so that vision is of little use for subterranean rodents here (Kott et al., 2010).” It would be more pertinent co cite Kott et al. 2014 (Journal of Zoology 294:68–76) in this context. Also exchange “here” by “there” in this sentence.
2. Lines 196 –197: “Furthermore, we did not find statistical sex-dependent differences in mean orientation for any of the solitary species.” Add statistics or explicit statement – statistics not shown.
3. Lines 197 – 199: “The ‘polygon method’ revealed that the alignment of the burrow systems of the Talas tuco-tuco deviated significantly from random orientation. A similar strong tendency was observable for the silvery mole-rat’s systems.” Refer to the Figure 1 here.
4. Lines 268 – 270: “However, the preferred mean axial directions of the burrow systems at the three localities did not differ statistically for both methods (Watson-Williams F-test applied), wherefore we pooled these data here (Figure 1).” I do recommend leaving out this argument. Because orientation was random at two out of three localities, mean directions of the burrow systems simply cannot differ statistically.
5. Lines 292–293: “….representing a highly interesting research question of heuristic potential.” Leave out “of heuristic potential”.
6. I did not go through the references systematically, but there seem to be a lot of inaccuracies in the reference list. For instance, ID numbers of articles are missing in some references (e.g., Hart et al. 2012; Lövy et al. 2015; Malkemper et al 2015); pages are not indicated in some references of book chapters (e.g. Hickman 1990; Lacey 2000; Moritz et al 2007).

---

## Round 0.3 · accepted · Accept

Thanks for making the additional amendments.

#